# Optimistic Verifiable Training by Controlling Hardware Nondeterminism

**Megha Srivastava**[*]
Department of Computer Science
Stanford University
`megha@cs.stanford.edu`

**Simran Arora**
Department of Computer Science
Stanford University
`simarora@stanford.edu`

**Dan Boneh**
Department of Computer Science
Stanford University
`dabo@cs.stanford.edu`

## Abstract

The increasing compute demands of AI systems have led to the emergence of services that train models on behalf of clients lacking necessary resources. However, ensuring correctness of training and guarding against potential training-time attacks, such as data poisoning and backdoors, poses challenges. Existing works on verifiable training largely fall into two classes: proof-based systems, which are difficult to scale, and "optimistic" methods that consider a third-party auditor who can replicate the training process and contest the trainer. A key challenge with the latter is that nondeterminism between GPU types during training prevents exact replication of the training process, resulting in schemes that are non-robust. We propose a method that combines training in a higher precision than the target, rounding after intermediate computations, and sharing rounding decisions based on an adaptive thresholding procedure, to successfully control for nondeterminism. Across three different NVIDIA GPUs (A40, Titan XP, RTX 2080 Ti), we achieve exact training replication at FP32 precision for both full-training and fine-tuning of ResNet-50 (23M) and GPT-2 (117M) models. Our verifiable training scheme significantly decreases the storage and time costs compared to proof-based systems, and is publicly released at `https://github.com/meghabyte/verifiable-training`.

## 1   Introduction

We are currently in the "large-scale era" of machine learning (ML), where the exciting capabilities of modern AI systems have required a dramatic increase in training compute needs [Sevilla et al., 2022]. In turn, several model training services, such as Replicate, OpenAI's Finetuning API, Together AI, Amazon Sagemaker, MosaicML Training, and Gensyn, have been created to support clients who lack the resources to train a model themselves. However, these services require clients to place a significant degree of trust in them to train the model correctly, without introducing a training-time attack such as data poisoning or undetectable backdoors [Wan et al., 2023, Goldwasser et al., 2022]. How can we help a client, such as an individual or a small company, hold the service provider accountable in case of misbehavior during training?

Consider an education start-up that wishes to finetune the Llama-70b language model (70B parameters) on their own curated dataset to support student learning. This task requires significant resources,

---

[*]Correspondence to `megha@cs.stanford.edu`.

38th Conference on Neural Information Processing Systems (NeurIPS 2024).

and the company might even lack the necessary expertise. Instead, they might choose to pay a $\mathrm{trainer}$ with vast computing resources to perform the training task (Figure 1). However, what if the $\mathrm{trainer}$ adds data points that spread misinformation, introduces a backdoor that advances a political agenda for specific prompts, or tries to save work by under-training the model? If the client starts to notice suspicious model behavior, is there any action they can take? We study this problem of *verifiable training*, or ensuring that the training of an ML model was performed correctly.

One possibility is for the $\mathrm{trainer}$ to provide the client with a cryptographic proof that the model was trained according to the specification. However, proof-based systems require cryptographic techniques that can be difficult to scale to the complexity of real-world ML systems. For instance, recent work based on zero-knowledge proof systems for verifiable *inference*, a much simpler task than training, requires more than 8 minutes to generate proofs for only 20 images [Liu et al., 2021]. Thus, practical proof-based methods for verifiable training have only been implemented for simple tasks such as logistic and linear regression [Garg et al., 2023, Ames et al., 2022].

An alternative "optimistic" approach is to consider a third-pary $\mathrm{auditor}$ (Figure 1). This could be a trusted 3rd party, such as a non-profit organization that may not have sufficient computing resources to provide training as a service beyond auditing, or a different provider that the client approaches and wishes to compare with the original model trainer. When a $\mathrm{client}$ suspects foul play, they can ask the $\mathrm{auditor}$ to challenge the $\mathrm{trainer}$ by training the model using the $\mathrm{auditor}$'s own compute, and demonstrate

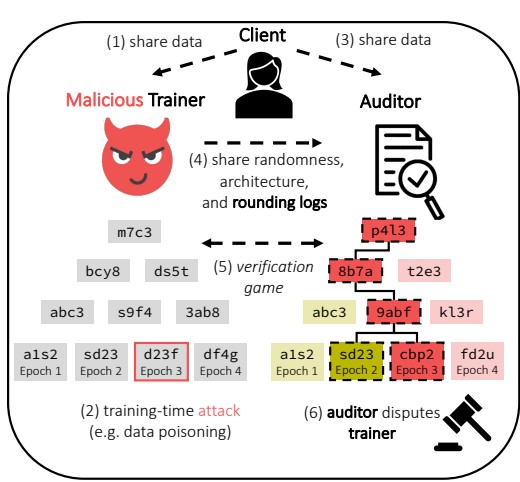

Figure 1: Overview of our scheme. After an $\mathrm{auditor}$ challenges a $\mathrm{trainer}$, they train the model, storing weights in a Merkle tree, and enter a binary search procedure to identify the exact steps of the dispute. We show how to control GPU nondeterminism between $\mathrm{auditor}$ and $\mathrm{trainer}$, expanding the set of potential auditors.

that the $\mathrm{trainer}$ did not train correctly. Based on the provided evidence required from the $\mathrm{auditor}$ (i.e. the precise timesteps model training diverged, as shown in Figure 1), the $\mathrm{client}$ can then choose to refuse the $\mathrm{trainer}$'s model, pursue legal action against the $\mathrm{trainer}$, or even dispute a potentially corrupt $\mathrm{auditor}$ if the client deems such evidence as incorrect, or another $\mathrm{auditor}$ disagrees. This protocol can be efficiently carried out using techniques from the literature on verifiable computing, such as the "verification game" method of Teutsch and Reitwießner [2019], which uses an interactive binary-search procedure to identify the exact intermediate computation step (e.g. training epoch) where the two parties diverged. Applying verifiable computation techniques to model training is particularly important given the increase in decentralized machine learning services like Gensyn, which seek to make ML compute more accessible by creating a network of many untrusted GPUs.

Unfortunately, the issue with such "optimistic" approaches is nondeterminism during model training: two models trained on different GPU types, even with same data order and random seed, can learn different weights (Figure 2). The $\mathrm{auditor}$ cannot simply compare their model weights with the $\mathrm{trainer}$'s, and recent work has shown that protocols based on comparing model weights, such as Jia et al. [2021]'s "proof of learning," are not robust and can be forged due to errors from nondeterminism [Thudi et al., 2022, Fang et al., 2023].

Our work addresses this limitation by asking: can the $\mathrm{trainer}$ provide additional information to the $\mathrm{auditor}$ that removes the effects of hardware nondeterminism? Our starting point is the observation that hardware nondeterminism occurs due to the accumulation of errors from floating point operations. For example, a matrix-vector multiply often results in different floating point values on different GPUs, since GPUs often accumulate in different orders. To address this issue, a natural approach is to perform training using a *higher* precision (e.g. FP32) than the target precision of the model weights (e.g. FP16), and periodically round back to the target precision. The hope is that all floating point errors will be confined to the higher precision bits, so that the rounded values *are* deterministic. However, this fails because computed values can occasionally straddle the "rounding boundary": i.e.,

the trainer can round up while the auditor rounds down, quickly causing them to diverge. Instead, we propose a solution where the trainer *records* the rounding direction for certain intermediate computation so that auditor can stay in sync with the trainer. As this requires the trainer to record a large number of bits, we also show how to reduce the amount of data needed to eliminate errors.

We use this strategy to adapt the verification game described by Teutsch and Reitwießner [2019] for verifiable training. The game's efficiency lies in our ability to store hashes of model checkpoints in a Merkle tree [Merkle, 1988]. To determine if training was performed according to the specification, the auditor needs to reconstruct the Merkle tree and compare the resulting Merkle root hash with the Merkle root hash provided by the trainer's – if they do not match, the two parties can enter an interactive binary search procedure to identify the exact training step of the dispute. The purpose of the binary search game is to hold both parties accountable: an auditor should not be able to simply claim that a model was improperly trained, but convince a third-party (e.g., the public, or a judge) by showing at what point during training the trainer misbehaved. We show our verifiable training scheme can scale to tasks such as full training of ResNet-50 (23M parameters) and finetuning of GPT-2 (117M parameters), significantly outperforming existing methods with respect to both time and storage cost, while eliminating statistical error due to non-determinism. For example, the proposal in prior work Jia et al. [2021] would require $> \mathbf{140\times}$ more storage cost than our method by comparing model weights at every step in order to achieve low (yet still non-zero) statistical error.

Concretely, our contributions include: (1) A method for two parties, training the same model on different GPU types, to achieve identical training results by logging and sharing rounding decisions; (2) A verifiable training scheme based on the verification game from Teutsch and Reitwießner [2019], which stores model weights in a Merkle tree for efficient comparison between a trainer and auditor; (3) Experiments showing the ability of our approach to scale to large models such as ResNet-50 and GPT-2 between three different NVIDIA GPU architectures (A40, Titan XP, RTX 2080 Ti); (4) Methods to reduce the storage cost of our approach via efficient encoding of rounding logs and an adaptive threshold mechanism to reduce the amount of rounding decisions logged; and (5) Comparisons with existing methods, including proof-based systems, that highlight the improved storage and time efficiency of our method. [2]

## 2    Related Works

Without any verifiable training scheme in place, significant trust is placed in the trainer, leaving a client vulnerable to many different attacks, such as the "poisoning" of data samples to cause undesirable behavior (e.g., generating unsafe text [Carlini et al., 2023, Koh et al., 2021, Wan et al., 2023]) and planting backdoors triggered by certain inputs [Goldwasser et al., 2022]. Therefore, training ML models in trusted environments has been an exciting direction explored by many researchers. One line of work consists of proof-based systems, where a proof of correctness (for a desired specification) is provided using cryptographic techniques such as succinct non-interactive arguments (SNARKs) [Micali, 1994, Bitansky et al., 2012, Lee et al., 2020, Liu et al., 2021, Garg et al., 2023, Kang et al., 2022]. However, even the most recent proof-based systems for verifiable training suffer extreme latencies, such as 22 minutes for training VGG-11 on one batch of 16 data inputs [Abbaszadeh et al., 2024], and have therefore primarily been developed for simpler models (e.g., logistic regression) that are less likely to be delegated to others in the first place [Garg et al., 2023, Ames et al., 2022]. Meanwhile, an alternative solution of training models in a trusted execution environment (TEE), such as NVIDIA's H100, incurs a performance penalty due to the cost of running inside a TEE [Dhanuskodi et al., 2023]. Furthermore, clients lose all security guarantees if an attacker can extract the attestation key from even one GPU [Nilsson et al., 2020, Bulck et al., 2018].

Our approach is most similar to proof-of-learning protocols, which consider a trusted 3rd party that compares checkpointing during the course of training with the original training sequence [Jia et al., 2021]. However, such methods not only incur high storage cost by requiring model weights to be stored frequently, but are non-robust due to errors from training nondeterminism. Several works have shown that proof-of-learning protocols can be spoofed and fail to verify correctness in several important contexts [Fang et al., 2023, Kong et al., 2023, Thudi et al., 2022]. Although Choi et al. [2023] recently proposed a verification procedure that is immune to several known proof-of-learning

---

[2]Our method is implemented entirely within the `pytorch` framework (compatible with version 2.3.1), and is available at `https://github.com/meghabyte/verifiable-training`.

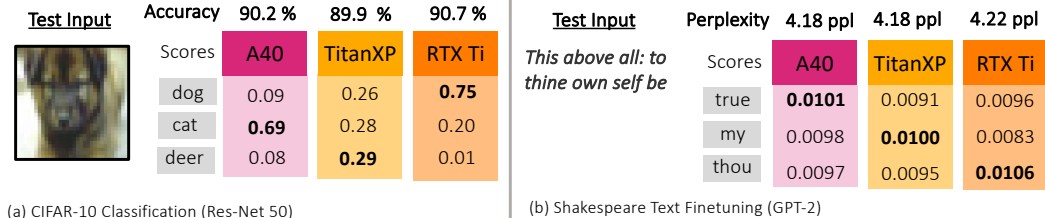

(a) CIFAR-10 Classification (Res-Net 50)    (b) Shakespeare Text Finetuning (GPT-2)

Figure 2: Even after ensuring the same software version, random seed, and use of deterministic algorithms via library flags, training nondeterminism persists between three GPU types.

attacks, their method is not only limited to supervised learning algorithms, but also based on an assumption that models temporarily overfit data during training, which may not always hold true.

**GPU Nondeterminism:** Prior work has investigated software patches for deterministic training, for instance by enforcing FP accumulation ordering, at a significant cost to efficiency Jooybar et al. [2013], Defour and Collange [2015], Chou et al. [2020], TensorFlow [2021], Zhuang et al. [2021]. While these options address deterministic computation on a *single* GPU architecture, achieving deterministic results across multiple GPU architectures remains challenging Crane [2018a], NVIDIA [2022]. We control hardware nondeterminism across GPUs in order to design an efficient and reliable verifiable training scheme. However, our method's impact extends beyond verifiable training, as training nondeterminism can have several negative consequences including bias, reproducibility, and downstream effects on ML pipelines [Zhuang et al., 2021, Crane, 2018b, Srivastava et al., 2020].

## 3  Set-Up: The Verification Game

Our method for verifiable training is based on the interactive verification game proposed by Teutsch and Reitwießner [2019] in the context of blockchains. The core idea is to resolve a dispute between a challenger, in our case the auditor, and a solver, in our case the trainer, for an expensive computation (e.g., model training). In order for the auditor to take any meaningful action (e.g., pursue legal action), they need to prove the exact source of the dispute (e.g., training time-step where an attack occurred). If we can save model weights at different time steps into a compact data structure such as a Merkle tree, then identifying the source of disagreement can be done efficiently using binary search [Merkle, 1988]. More precisely, the verification game consists of the following parties:

1. trainer, who has putatively trained a model according to a client's specifications. In our example, this is a service provider with sufficient compute power to train a model.
2. client, who receives a model from the trainer and approaches an auditor.
3. auditor, who officially challenges the trainer on behalf of a client. This is a trusted 3rd-party that has sufficient resources but does not necessarily provide training as a service. The client can choose several auditors to audit the trainer's model.
4. judge: Sometimes a judge may need to arbitrate a legal claim. The judge can only perform minimal computations (e.g., one training epoch), but can examine the auditor's claims and enforce a penalty against either the trainer, for incorrect training, or the auditor, for a false alarm.

When the trainer is approached by an auditor, they would need to share training parameters, model architecture, and randomness, as shown in Figure 1. The auditor would then replicate the training process, storing model weights in a Merkle tree at the same checkpointing interval as the trainer (every leaf node in a Merkle tree is a hash of the data and every non-leaf node is a hash of its children). The main loop of the verification game starts when both parties have the root of their respective Merkle trees. If training was performed correctly, then the trainer's root should match the auditor's. Otherwise, a binary search procedure is performed, where the auditor iteratively descends the Merkle tree until it identifies two consecutive leaf nodes, $i$ and $i + 1$, where the hash at $i$ matches that of the trainer, but the hash at leaf $i + 1$ does not. This identifies the point in the computation of the dispute.

This interactive verification game requires the cooperation of the trainer. If the trainer refuses to share the value at a certain node of their Merkle tree within a given time frame, they can be considered to have failed the audit. Additionally, the trainer and auditor use a Merkle tree to store model weights, requiring far less storage than prior work, if correct training produces identical weights (and

| Example | Sum Order | FP32 | Rounded to FP16 |
|---|---|---|---|
| $a, b, c = 0.1, -0.1, 0.2$ | $a + b + c$ | 0011111001001100110011001101 | 0011001001100110 |
| | $a + c + b$ | 0011111001001100110011001110 | 0011001001100110 |
| $a, b, c = 10.02, 13.162813186645508, 0.2$ | $a + b + c$ | 01000001101110110001000000000001 | 0100110111011001 |
| | $a + c + b$ | 01000001101110110001000000000000 | 0100110111011000 |

Table 1: Two examples of floating point accumulation error when rounding arithmetic performed higher precision (e.g. FP32) down to lower precision (e.g. FP16). In the second example, the error in the FP32 result transfers to the rounded FP16 result.

identical hash values).The problem is that training nondeterminism leads to weight divergence, and causes this verification game to always fail. This why we seek to prevent divergence in training.

# 4 The Nondeterminism Challenge

Although there are user-side controls for forcing deterministic operations within a single GPU architecture , these controls do not prevent nondeterminism between GPU architectures (e.g., NVIDIA H100 and V100), where trained models can have similar aggregate performance (e.g., accuracy) yet yield very different predictions, as shown in Figure 2 Crane [2018a], NVIDIA [2022]. There are three main sources of nondeterminism between GPU types:

**1. Floating-Point Arithmetic:** Computers represent real values using integer and FP representations, typically the IEEE 754 standard (Figure 5). There is a tradeoff between the approximation fidelity and the # of bits used to represent the real values. The chosen precision controls the representable numerical range (e.g., 32-bit FP values can represent values between $1.17549435e - 38$ and $3.40282347e + 38$). Because computers round to representable FP values, changing the order in which FP values are accumulated can change the resulting sum Kahan [1965], Whitehead and Fit-Florea [2011]. Over the course of the many operations during training, this can lead to a large difference in the end result between the trainer and auditor.

**2. Parallel Computation:** In a GPU, a single operation (called a *kernel*) is executed by thousands of threads in parallel. GPUs contain a set of *streaming multiprocessors* (SMs), which run the *thread blocks* required for the kernel. At the hardware level, these blocks are divided into *warps* that are assigned to the available cores. Because different GPUs have a different number and size of compute units, applications partition arithmetic workloads (e.g., batch matrix multiplies) differently to achieve high performance NVIDIA [2022], thus changing the order of FP operations.

**3. Memory Hierarchy and Variable Delays:** The time taken for memory access by each thread depends on the physical location of the data, which can create variable delays Jooybar et al. [2013], Defour and Collange [2015], Chou et al. [2020]. The GPU memory hierarchy consists of large amounts of high bandwidth memory (HBM) and small amounts of fast SRAM memory, and maintains an L1 and L2 cache to improve access times. The caches sizes and access times differ across GPU architectures (e.g. an NVIDIA A100 has 192KB / 40 MB of L1/L2 cache memory, while the H100 has 256KB / 50MB). This affects warp scheduling, leading to changes in operation ordering resulting in nondeterminism. Finally, to compute primitives such as GEMMs ($D = A \cdot B + C$), the workhorse of machine learning, GPUs split the work of computing the tiles of $D$ across a thread block NVIDIA [2023], resulting in nondeterminism that a robust verification method needs to control.

# 5 Method Overview

## 5.1 Accumulation Errors Start at Higher Precision Bits

Our key idea is that if nondeterminism of training between GPU types occurs due to FP operations, then any error will initially be introduced in the lower bits. Suppose that both trainer and auditor train at a *higher* FP (e.g., $b_{tr} = 64$) precision than the client's target model precision (e.g., $b_m = 32$) and then periodically *round* (e.g., $b_r = 32$) after intermediate computation steps (e.g., a convolution layer). One might hope that this will "erase" the errors due to nondeterminism, and prevent them from accumulating. Unfortunately, simply rounding to the nearest FP32 after each computation during training is insufficient for determinism. The problem is due to rounding errors that straddle

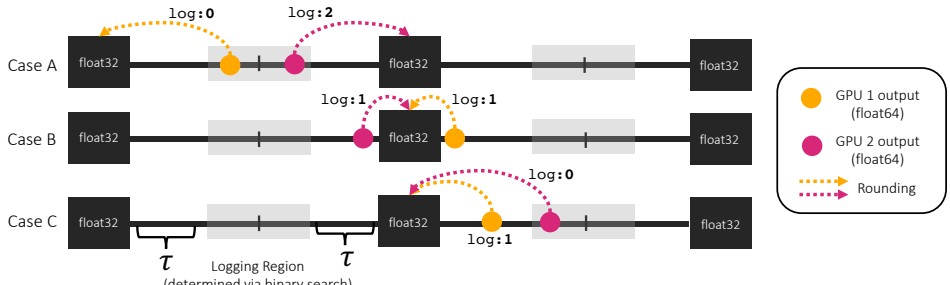

Figure 3: Divergence between outputs on two different GPUs (in FP64) for a given function and input can result in different rounding choices when rounding to the nearest FP32. We only wish to log rounding decisions for Case A, where the auditor should copy the trainer's rounding choice in order to reach the same value. This requires defining a logging region, determined by a threshold $\tau$,

the *rounding boundary*. Consider Case A in Figure 3, which shows a divergence in the output of a computation using FP64 on two different GPUs. Because the outputs of GPU 1 and 2 are on different sides of the boundary, rounding to the nearest FP32 results in different values, introducing error.

What if the trainer records their rounding choice (e.g., up, down, none) for every intermediate computation? The auditor could then copy the trainer's choice, and therefore round to the exact same value and successfully control for nondeterminism. However, the auditor should not copy the trainer's behavior for every output (see Cases B & C, Figure 3). If a computation output on GPU 1 is too close to the rounded value, then it is possible that GPU 2 is also close in distance but from the opposite direction. In this case, the auditor should ignore the trainer's choice. We therefore need to introduce a threshold $\tau$ under which the trainer does not record their rounding choice.

Our method requires upper bounding the divergence $d_{div}$ between any two different GPUs for any intermediate computation $f$ (i.e. difference in outputs for the same input). Let $\epsilon_b$ represent the distance between two FP32 values, after rounding to $b_r$ bits of the mantissa (Figure 5) and controlling for the exponent. We need to select $b_r$ and $\tau$ such that $d_{div} < \epsilon_{b_r}$ and $d_{div} < 2\tau$ (Figure 3). Because the set of possible FP numbers is finite, there exist optimal bounds for $b_r$ and $\tau$. In practice, we find that $b_r \leq 32$ and $\tau > 0.25 \cdot \epsilon_{32}$ are sufficient for standard intermediate computations in neural network training (e.g., convolution, layer norm) in FP64. We study different values for $b_r$ in Section 6.

## 5.2 Primitives

We assume both trainer and auditor train models using the IEEE-754 standard FP numbers (Figure 5). Besides requiring read and write disk I/O operations, we define the following functions:

1. $\mathsf{rnd}_{b_r}(x)$: rounds input $x$ to the nearest FP up to $b_r$ bits of the mantissa, as shown in Figure 5.
2. $\log(x, b_r, \tau, f)$: logs to file $f$ a logging direction $c$, which is either 0 (down), 1 (ignore), or 2 (up) depending on threshold $\tau$ and rounding amount $b_r$, as shown in Algorithm 4.
3. $\mathsf{rev}(x, b_r, c)$: reverses rounding of input $x$ based on logging direction $c$. If $x < \mathsf{rnd}_{b_r}(x)$ & $c = 0$, then return $x$ rounded to the nearest float *below* $x$ with $b_r$ precision. If $x > \mathsf{rnd}_{b_r}(x)$ & $c = 2$, then return $x$ rounded to the nearest float *above* $x$ with $b_r$ precision. Otherwise, do not correct.
4. $\mathsf{threshold}(l, b_r, b_{tr})$: identifies the optimal threshold to log rounding directions (0 or 2) instead of 1, which the rev function ignores, based on the binary search procedure in Section 5.4.
5. $\mathsf{hash}_{\mathsf{sha256}}(\theta)$: creates a SHA-256 hash of provided model weights $\theta$ (in $b_m$ precision).
6. $\mathsf{tree}(leaf_1, leaf_2 ..., leaf_n)$ : create a Merkle tree where each *leaf* node is the output of $\mathsf{hash}_{\mathsf{sha256}}(\theta)$ for model weights $\theta$ at a given checkpoint, with a checkpointing interval $k$ [Merkle, 1988].

## 5.3 Training and Auditing

The trainer's task begins when a client approaches them with dataset $D$, training specifications (epochs $E$, loss function loss, etc.), and a requested model precision $b_m$. The trainer can then choose a training precision $b_{tr} > b_m$, a rounding amount $b_r \leq b_m$, and a checkpointing interval $k$ to periodically store small $\mathsf{hash}_{\mathsf{sha256}}(\theta)$ of model weights $\theta$ in a Merkle tree, for efficient comparison with an eventual auditor. Then, as detailed in Algorithm 1, the trainer can perform training as normal, but after every intermediate computation (e.g., convolution) perform the $\mathsf{rnd}_{b_r}$ operation on

each output. Rounding is applied to computations in both the forward and backward passes. Finally, either using a fixed threshold $\tau$ or a layer-specific optimal $\tau$ from the threshold function described in Section 5.4, the trainer applies log, which logs rounding choices *only for the computations an auditor should copy*. The output of the algorithm includes a rounding log file $F$ and the root of the Merkle tree which, along with the shared randomness R and all training parameters, the trainer can share with any trusted third-pary auditor who challenges them.

After a client approaches them, the auditor initiates the verification game described in Section 3. To avoid penalty, the trainer must cooperate by sharing the rounding amount $b_r$, randomness R used in training (e.g., a pseudo-random number generator), the checkpointing interval $k$, and set of rounding logs $F$. The auditor then follows the training procedure and corrects their rounding choice (e.g., up or down) to match those logged in $F$ using the rev operation, as detailed in Algorithm 2 (Appendix). By correcting each rounding mismatch during the course of training, the auditor is able to prevent nondeterminism errors from accumulating. Therefore, the auditor can store the $\mathsf{hash}_{\mathsf{sha256}}(\theta)$ of model weights $\theta$ in a Merkle tree at interval $k$, knowing that if training was done correctly, the model weights should be identical to the trainer's at any timestep. The output of Algorithm 2 is the root of the auditor's Merkle tree, which they can use to compare with the trainer's root.

## 5.4 Reducing storage cost

Logging rounding decisions for every neural network layer output during training incurs a large baseline storage cost, and is our main limitation. For dataset $D$, batch size $B$, training epochs $E$, and model layers $L_\theta$, the upper bound on the total storage cost for verifiable training with our method is:

$$\text{storage cost (B)} = |D| \times E \times B \times (\sum_{l=1}^{L} o_{l,f} + \sum_{l=1}^{L} o_{l,b})$$

(1)

where $o_{l,f}$ and $o_{l,f}$ represent the size of outputs of the forward pass and backward pass of layer $l$. Note that the log entries do not need to be kept around in the RAM and can be written straight to the disk. Moreover, this cost is a one-time cost incurred by the trainer, who in our context is likely to be a powerful commercial provider with access to such storage capacity. Furthermore, as we later show in Section 6, for models with many linear layers like Transformer-based language models (e.g., GPT-2), where parameters significantly outnumber intermediate computations, this storage cost is significantly smaller than alternative approaches that require saving model weights [Jia et al., 2021]. Nevertheless, we now describe our method for reducing storage cost by (i) efficiently encoding rounding logs and (ii) adaptive selection of the threshold $\tau$ to reduce the storage costs.

**Efficient Encoding:** Each log entry is a value from the set $0, 1, 2$, as opposed to the FP model weights. We pack sub-sequences of five log entries into a single byte via a fast GPU-based radix-3 to radix-2 conversion, yielding 1.6 bits/entry storage that is close to the best possible packing of 1.58 bits/entry, and yields a 77% storage reduction relative to naively storing one log entry per byte.

**Adaptive Threshold:** Recall that we need to select a threshold $\tau$ that controls for whether the trainer logs a rounding choice, or instead logs 1 which the auditor ignores. The more one can increase $\tau$, the more 1 values are recorded, which can make rounding logs more compressible (due to long sequences of 1s). Furthermore, it is possible that the divergence $d_{div}$ between outputs on two different GPUs, given the same input, is function-specific. For example, while convolution requires several matrix multiplications that might result in a large FP accumulation error, normalization operations are unlikely to result in large $d_{div}$, and a larger $\tau$ can be applied. We develop an efficient algorithm (Algorithm 3 in the Appendix) to find the optimal value for $\tau$ given a particular layer and data of output values that led to different rounding choices between any two GPUs (e.g., Case A in Figure 3). For a given rounding amount $b_r$ and training precision $b_{tr}$, the algorithm performs a binary search between $\tau = 0.25 \cdot \epsilon_{32}$ (our upper bound on the $d_{div}$ between two GPUs for any function) and $\tau = 0.5 \cdot \epsilon_{b_r}$ (the rounding boundary). By performing this procedure for the different intermediate computations in a model, the trainer can hope to better compress the rounding log $F$.

**Merkle Tree Storage:** Storing SHA-256 hashes of model weights during training in a Merkle tree creates an efficient mechanism for the verification game described in Section 3, with negligible storage requirements. The audit ends when either the trainer withdraws, the auditor confirms that training was performed correctly, or the auditor can present paths to the two leaves of their Merkle tree where divergence starts, providing evidence to dispute the trainer.

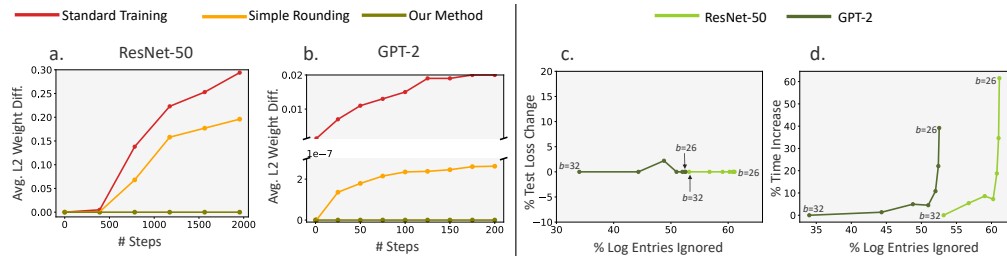

Figure 4: We successfully control for nondeterminism between GPU types for both ResNet-50 (a.) and GPT-2 (b.) tasks, while standard training and simple rounding without performing rev corrections result in model divergence over the course of training. Stronger rounding has minimal affect to model performance (c.), but at the cost of increasing time for trainer (d.).

Table 2: Efficient encoding reduces storage requirements by 77%, and rounding to $b = 26$ improves the compression further between 5-20% (values reported for 1 step of training). The original proof-of–learning protocol from Jia et al. [2021] requires storing 2.78 GB of model weights for GPT-2, or more than **140x** our storage cost, while still incurring statistical error.

|  | **ResNet-50** $b = 32$ | **ResNet-50** $b = 26$ | **GPT-2** $b = 32$ | **GPT-2** $b = 26$ |
|---|---|---|---|---|
| Naive Encoding | 456 MB | 456 MB | 92 MB | 92 MB |
| Efficient Encoding | 105 MB | 105 MB | 22 MB | 22 MB |
| + Zip Compression | 96 MB | 91 MB | 20 MB | 18 MB |

# 6 Empirical Results

We evaluate our verifiable training method on the two large-scale models listed below with all possible trainer and auditor pairs across NVIDIA GPUs A40, TITAN Xp, and RTX 2080 Ti (see Appendix B for more details). In Section 6.2, we compare our method with recent proof-based systems.

1. **ResNet-50**: We train (from random initialization) ResNet-50 (23M) on CIFAR-10 with dataset size 50K & batch size B=64. Test accuracy = 90.7% after 100 epochs training on Titan RTX Ti.
2. **GPT-2**: We finetune GPT-2 (117M) on a corpus of Shakespeare text with dataset size 1.1M tokens, batch size B=8, and sequence length 64. Perplexity = 4.22 after 1 epoch training on Titan RTX Ti.

Figure 2 shows that nondeterminism due to GPU architecture exists for both tasks. While we can repeatedly obtain identical results across training runs on the same GPU architecture, training on different GPU architectures results in fundamentally different models.

## 6.1 Implementation and Findings

We implement our verifiable training method entirely on top of the `pytorch` framework, with `torch` version 1.13.1 and CUDA version 11.7. The intermediate computations we apply $rnd_b$ to are layers (e.g., `torch.nn.Conv2D`) in the model's computation graph. Rounding-related operations (rnd and rev) either using casting or FP functions (e.g., `torch.nextafter`) that can run on the GPU, thus having little impact on computational speed. Because we observed that the `torch.randn` operation used for dropout in GPT-2 is non-deterministic for long inputs (even for the same seed, see Appendix I), we implement our own dropout as our method requires shared randomness $R$.

**Successful control for non-determinism:** Our method completely eliminates non-determinism between full training runs of both for both the ResNet-50 training and GPT-2 fine-tuning tasks across all possible trainer and auditor pairs between the A40, Titan XP, and RTX 2080 Ti GPUs. As Figure 4 shows, standard FP32 training results in an increasing divergence (l2-distance of weights) between models on different GPUs over the course of training. Furthermore, we show the simple approach of training in FP64 and rounding to FP32 after every intermediate computation, but without the auditor correcting rounding decisions with rev, fails to mitigate this issue. Only our method, in which the auditor follows the rounding decisions ($b_r = 32$) made by the trainer for every intermediate computation, eliminates non-determinism and persists over time. Our implementation, which requires disk I/O during training to store the rounding decisions, results in a small increase in training time for the trainer (1.2-1.4x) and auditor (1.3-1.7x) using a non-optimized, prototype implementation (Table 5). We report the storage requirements of our method in Table 2, showing

Table 3: Average # of rev corrections performed by auditor per training step. Even at $b = 32$, auditing only requires 20-25 corrections (**2e-6 to 9e-6%** of samples) per training step.

| **ResNet-50** | $b = 32$ | $b = 31$ | $b = 30$ | $b = 29$ | $b = 28$ | $b = 27$ | $b = 26$ |
|---|---|---|---|---|---|---|---|
| Forward | $15 \pm 3$ | $6 \pm 2$ | $3 \pm 1$ | $3 \pm 1$ | $0$ | $0$ | $0$ |
| Backward | $10 \pm 0.6$ | $6 \pm 0.6$ | $2 \pm 1$ | $0.7 \pm 0.7$ | $0 \pm 0$ | $0 \pm 0$ | $0 \pm 0$ |
| **GPT-2** | $b = 32$ | $b = 31$ | $b = 30$ | $b = 29$ | $b = 28$ | $b = 27$ | $b = 26$ |
| Forward | $2 \pm 0.7$ | $2.3 \pm 0.8$ | $2.2 \pm 0.4$ | $0.2 \pm 0.2$ | $0.4 \pm 0.2$ | $0 \pm 0$ | $0 \pm 0$ |
| Backward | $19 \pm 13$ | $0.75 \pm 0.3$ | $1.2 \pm 0.4$ | $0.2 \pm 0.2$ | $0. \pm 0.0$ | $0 \pm 0$ | $0 \pm 0$ |

Table 4: Adaptive thresholds identified for different operations using Algorithm 3 with $b = 32$.

| | **2D Convolution** | **Batch Norm** | **Linear** | **Layer Norm** |
|---|---|---|---|---|
| Dimension | *256 (1,1)* | *(128, 128, 16, 16)* | *(768,768)* | *(768,1)* |
| $\tau$ | $0.305 * 2^{-23}$ | $0.499 * 2^{-23}$ | $0.465 * 2^{-23}$ | $0.499 * 2^{-23}$ |

that our efficient encoding scheme reduces the size of the trainer's rounding logs by 77%, relative to naive logging. Because the Merkle tree stores 32-byte SHA-256 hashes, its overall size (KBs) and creation time are negligible and not reported. Finally, we show that decreasing the rounding amount $b$ to values even as low as 26 has little effect on model performance (we observe no change in accuracy, so report test loss), but increase training time (Figure 4). We observe that smaller values of $b$ do allow more log entries to be ignored, improving compression of the file, which we discuss next.

**Compression with adaptive threshold:** Our approach outperforms (Table 2) the storage costs of proof-of-learning protocols that save model weights for GPT-2 (2.78GB), which has many linear layers – we observe more than **140x** reduction relative to the approach in Jia et al. [2021]. We further reduce the storage cost of our method by decreasing the rounding amount $b$ and implementing the adaptive thresholding strategy (Section 5.4). Table 4 reports adaptive thresholds $\tau$ for four different pytorch layers at rounding amount $b_r = 32$. Convolutions require the lowest $\tau$, indicating larger divergence in outputs between GPU types, which is expected due to the large # of matrix multiplications. Meanwhile, $\tau$ is higher for normalization layers, likely due to smaller divergences between GPU types. Because adaptive thresholding seeks to reduce the # of times rounding decisions (0 and 2) are logged and improve log file compression, we report storage cost after zip compression in Table 2. As expected, more aggressive rounding (which results in a higher $\tau$) improves the compression rate. Although the compression gains are mild in comparison to our encoding step, they build-up over the course of training. Finally, we report the average # of rev corrections an auditor needs to perform for one training step in our two tasks (Table 3). These values are surprisingly small in comparison to the # of operations logged – only a maximum of **2e-6%** (ResNet-50) and **9e-6%** (GPT-2) of logged values, are actually needed by the auditor! We also observe that severe rounding (e.g., $b = 27$) completely eliminated the hardware non-determinism for our tasks, requiring no corrections from the auditor. This shows a huge gap between the # of values currently saved by the trainer and those needed by the auditor, motivating an exciting future possibility of significantly reducing the storage cost of our method if we could reliably predict when a divergence will not occur.

## 6.2 Comparison with alternative approaches

**Logistic Regression:** Garg et al. [2023] recently proposed a zero-knowledge proof-based system for verifiable training of a logistic regression, which importantly does not leak information about the client's data or require a trusted third-party auditor, unlike our work. However, since verifiable training itself is motivated by a client not having sufficient resources to train the model, it is crucial to consider the implications of scale. The authors report the prover time and proof size requirements for one training pass of logistic regression on a dataset of $2^{18}$ items, with 1024 dimensions and a batch size of 2014, as **72 seconds** (training and proof generation time) and **350 MB** respectively. We replicate this training task, and find that our method significantly improves upon both storage and time requirements, requiring only **106 KB** and **7 seconds** (both training and auditing). Furthermore, because Garg et al. [2023] do not report the duration of "offline phase" of their method, their reported value is a lower bound on the actual time required. Finally, we note that the original proof-of-learning protocol from Jia et al. [2021], which also considers a trusted third-party, would require **9.2 MB per training step** to store all model weights. Therefore, our method is at least **85x** more space efficient.

**VGG-11:** Concurrent to this work, Abbaszadeh et al. [2024] introduce a zero-knowledge proof-of-training protocol for deep neural networks, presenting results for one batch step of training for

a simplified version of the VGG-11 model with 10M parameters, which is less than the original VGG-11 network and ResNet-50 [Simonyan and Zisserman, 2015]. While the authors do not provide architectural details, we can assume that increasing the # of parameters to the original VGG-11 would only increase their reported proof time and size. We, therefore, compare their reported values with an implementation of our method for the same task of verifying the training of VGG-11 on CIFAR-10 with a batch size of 16. While their use of incrementally verifiable computation leads to tractable proof size (1.36MB vs. the 1.2MB per iteration cost of our method), Abbaszadeh et al. [2024]'s method requires **22 min. per training iteration**. In comparison, our method requires training and auditing times of only 6 sec. per iteration and is significantly more efficient (factor of **220x**), an important consideration for model training as a commercial service.

Finally, in Appendix Section J, we compare our results with an adaption of Gupta et al. [2023]'s protocol for secure inference of GPT-2. Compared with our method's storage cost (18MB) and training time (11s for training, 13.5s for auditing), scaling Gupta et al. [2023]'s protocol for training would introduce around a **10,000x** data and **40x** time overhead. While proof-based systems provide strong security guarantees without a third party, they do so at the cost of relying on hard-to-scale cryptographic techniques, as well as approximating non-linear functions that can harm performance.

# 7 Security Analysis

Our work makes a *1-of-n* honesty assumption, i.e., as long as one of $n$ auditors is honest, any attack from a malicious trainer that results in diverging model weights will be detected. One consideration is the potential manipulation of the rounding logs by an adversarial trainer who could select rounding decisions that achieve a desired outcome, and which the auditor would follow. Concretely, let us define our threat model so that the trainer knows an auditor's GPU a priori. Recall that an auditor only copies the trainer's rounding decision in Case A in Figure 3, when both GPUs compute values close to the rounding boundary. Under this threat model, the trainer can identify the $n$ steps where the auditor is close to the boundary (as in Case A), enumerate the set of $2^n$ different models that result from different rounding decisions, and selectively pick a model that exhibits a desired property.

However, the trainer cannot use this strategy to embed an arbitrary property (e.g., a specific backdoor). It can only select from the set of models that differ in certain rounding decisions, which all require the trainer to use the correct training specifications accepted by the client (such as exact training data & hyperparameters). Furthermore, since the expected # of divergences between the trainer and the auditor is extremely small (see Table 3), the set of possible models where an auditor would not detect an attack (e.g., many rev ops) is limited. Finally, we show in Table 6 in the appendix that the divergence (measured both as $\ell_2$-norm between model weights and output distributions) due to GPU non-determinism is significantly less than the divergence due to data ordering during training. Therefore, if a client will accept a model trained with *any* random ordering of the data during training, then it is unlikely that an adversarial trainer — that can only alter rounding decisions — could produce a model that the client would not accept. Nevertheless, fully understanding the model properties obtained by manipulating rounding logs adversarially is an important future direction.

# 8 Limitations and Future Work

Our verifiable training scheme successfully controls for hardware nondeterminism. It expands the pool of potential auditors of a model training service, allowing us to envision a world where a client can even use two competing service providers it trusts to audit each other. Relative to proof-based systems, a limitation is the need for all parties to trust the third-party auditor. If the trainer provides finetuning services on top of closed-source models (e.g., OpenAI), then our scheme will only work for the third-party auditors that the trainer is willing to share model weights with. Other limitations included the added latency of training in higher precision and the storage cost. While we have shown that our method requires significantly less storage than alternatives, the vast majority of stored rounding decisions are not used by the auditor and are therefore unnecessary (Section 6). Therefore, an exciting direction for future work is to mitigate this gap by better predicting when GPU divergence between computations occurs. Recent work has similarly argued for a stronger profile of noise during training in the context of verification [Fang et al., 2023]. Finally, another direction for future work includes adapting our method for distributed training [Li et al., 2020].

# 9 Acknowledgements

We thank Bill Dally, Duncan Riach, Gabriel Poesia, and Chris Ré for helpful discussion and feedback. Megha Srivastava was supported by an IBM PhD Fellowship and the NSF Graduate Research Fellowship Program under Grant No. DGE-1656518. In addition, this work was funded by NSF, DARPA, the Simons Foundation, UBRI, and NTT Research. Opinions, findings, and conclusions or recommendations expressed in this material are those of the authors and do not necessarily reflect the views of DARPA.

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

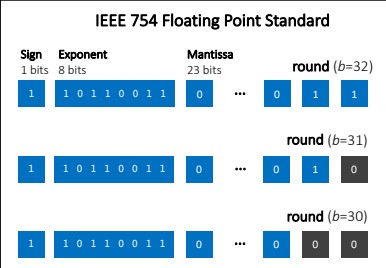

Figure 5: We define rounding to $b$ bits as rounding to the nearest 32-bit FP number that has 0s in the last $32 - b$ bits of the mantissa, after accounting for the exponent.

Nathan Whitehead and Alex Fit-Florea. Precision & performance: Floating point and ieee 754 compliance for nvidia gpus, 2011. URL `https://developer.nvidia.com/sites/default/files/akamai/cuda/files/NVIDIA-CUDA-Floating-Point.pdf`.

NVIDIA. Cuda: Hopper tuning guide, 2023. URL `https://docs.nvidia.com/cuda/pdf/Hopper_Tuning_Guide.pdf`.

Karen Simonyan and Andrew Zisserman. Very deep convolutional networks for large-scale image recognition, 2015.

Kanav Gupta, Neha Jawalkar, Ananta Mukherjee, Nishanth Chandran, Divya Gupta, Ashish Panwar, and Rahul Sharma. Sigma: Secure gpt inference with function secret sharing. Cryptology ePrint Archive, Paper 2023/1269, 2023. URL `https://eprint.iacr.org/2023/1269`. `https://eprint.iacr.org/2023/1269`.

Shen Li, Yanli Zhao, Rohan Varma, Omkar Salpekar, Pieter Noordhuis, Teng Li, Adam Paszke, Jeff Smith, Brian Vaughan, Pritam Damania, and Soumith Chintala. Pytorch distributed: experiences on accelerating data parallel training. *Proc. VLDB Endow.*, 13(12):3005–3018, August 2020. ISSN 2150-8097. doi: 10.14778/3415478.3415530. URL `https://doi.org/10.14778/3415478.3415530`.

## A   IEEE Floating Point Image

See Figure 5.

## B   GPU Details

All experiments reported in our paper are run with the following three GPUs:

- NVIDIA Titan XP: 3840 Cores, 12 GB
- NVIDIA RTX 2080 Ti: 4352 Cores, 11 GB
- NVIDIA A40: 10752 Cores, 48 GB

We are able to successfully replicate training runs between all pairs of these 3 GPUs.

## C   Logging Algorithm

See Algorithm 4

## D   Train Algorithm

See Algorithm 1.

## E   Audit Algorithm

See Algorithm 2.

## F   Adaptive Thresholding Algorithm

See Algorithm 3.

## G   Time Requirements

See Table 5.

## H   Model Divergence Comparison

See Table 6.

## I   Random Number Generation

Our verifiable training scheme requires shared randomness between the trainer and auditor, which is used for deciding input data batching, weight initialization, and operations such as dropout (randomly setting outputs to zero). More formally, our scheme requires sharing the same random seed and pseudo-random generator. However, in our implementation based on pytorch (assuming the same software version between trainer and auditor), we chose to rely on the the torch random seed functionality. While this successfully controls for batch input ordering and weight initialization, it is unfortunately not sufficient for random number generation, as operations such as torch.nn.randn() leverage parallelism when the requested # of values is higher than a certain amount. Specifically, we found that across T40, RTX 2080 Ti, V100, A40, and A100, given the same seed, torch.randint() produces identical tensors onlt up to size 40960. At size 40961, T40 (which is an older GPU) deviated from the rest. Likewise, at size 69633, 2080 Ti deviated from the rest, and so on. Based on these observations, we arranged for calls to torch.randint() in the dropout layer (which is the only operation using large random tensors in our tasks) to be replaced by generating and concatenating multiple random tensors of size 40960 or less. Specifically, a random tensor of size $n > 40960$ is generated by concatenating $(n//40960)$ random tensors of size 40960 and one random tensor of size $(n\%40960)$. However, we emphasize that it is therefore important in our scheme either for both parties to implement this change a priori, or simply use an external source for pseudorandomness.

## J   Comparison with GPT-2 Inference

The previously discussed proof-based systems for verifiable training by-pass the need for a third-party auditor, but very few efficient systems exist in the literature. Many more works study secure *inference* of deep neural networks, which could be used to construct verifiable training protocols with stronger security guarantees than ours (e.g., allowing a trainer to keep a proprietary model's weights private), but come at a significant cost to performance and resources. To demonstrate this, we consider adapting Gupta et al. [2023]'s protocol for secure inference of GPT-2 based on multi-party computation, to our context of verifiable training. Gupta et al. [2023] show how two parties, the client with private data and the trainer, can jointly compute the forward pass of a known model architecture without revealing additional information beyond the model output to each other. Because they report the the communication overhead $P = 0.37$GB and time $T = 0.96$ seconds for one forward pass on a single data input, we can calculate $2 \times P \times D \times E = $ **189 GB** and $2 \times T \times D \times E = $ **983 seconds** as estimated communication cost and time, respectively, for 1 step of training in out GPT-2 task, where 2 considers both the forward and backward pass. Compared with our method's required storage cost (18MB) and training time (11s for training, 13.5 seconds for auditing), scaling Gupta et al. [2023]'s protocol for training would introduce around a **10,000x** data and **40x** time overhead.

---

**Algorithm 1** train

---

INPUT: dataset $D$, epochs $E$, batch size $B$, shared randomness $R$, model $W_\theta$, loss function loss, rounding amount $b_r$, training precision $b_{tr}$, target model precision $b_m$, checkpointing interval $k$
OUTPUT: Merkle tree root $M_{root}$, rounding log file $F$

1: $F, M_{leaves} \leftarrow$ create empty file and leaf list
2: $W_\theta \leftarrow \mathsf{init}(R, b_{tr})$ //initialize weights
3: $T \leftarrow \frac{D*E}{B}$
4: **for** $t = 1...T$ **do**
5:    $input \leftarrow \mathsf{batch}(R, D, B)$ // get data batch
   // forward pass
6:    **for** layer $l_\theta \in W_\theta.$layers **do**
7:      $output \leftarrow l_\theta(input)$
8:      $\tau \leftarrow \mathsf{threshold}(l_\theta, b_r, b_{tr})$ //set threshold
9:      $\mathsf{log}(output, b_r, \tau, F)$
10:     $output \leftarrow \mathsf{rnd}_{b_r}(output)$
11:     $input \leftarrow output$
12:   **end for**
13:   $loss \leftarrow \mathsf{loss}(output)$
14:   // backward pass, reversed layers
15:   $grad\_output \leftarrow \nabla_{\mathsf{loss}}$
16:   **for** layer $l_\theta \in W_\theta.$layers **do**
17:     $grad\_input \leftarrow \nabla_{l_\theta}(grad\_output)$
18:     $\tau \leftarrow \mathsf{threshold}(\nabla_{l_\theta}, b_r, b_{tr})$
19:     $\mathsf{log}(grad\_input, b_r, \tau, F)$
20:     $grad\_input \leftarrow \mathsf{rnd}_{b_r}(grad\_input)$
21:     $grad\_output \leftarrow grad\_input$
22:   **end for**
23:   $\theta \leftarrow$ update update weights
24:   **if** $t \bmod k = 0$ **then**
25:     $M_{leaves}.$append$(\mathsf{hash}_{\mathsf{sha256}}(\theta$ in precision $b_m))$
26:   **end if**
27: **end for**
28: $M_{root} \leftarrow \mathsf{tree}(M_{leaves})$ // create Merkle tree
29: **return** $F, M_{root}$, and model $W_\theta$ in target precision $b_m$

---

Table 5: Training time requirements, including Merkle tree operations (at $k = 5$), for 1 step of training broken down by stage of our verifiable training process. Note that reported times are specific to the particular dataset, batch size, and task, and using a non-optimized prototype codebase – therefore the relative increase is time is more important.

|  | ResNet-50 | GPT-2 |
|---|---|---|
| Original (No Rounding or Disk I/O) | 24s | 8s |
| Trainer | 28s | 11s |
| Auditor | 31s | 13.5 |

Table 6: Comparison of model divergence due to data ordering versus GPU non-determinism. Reported numbers are averaged between 10 pairs of models, error bars are standard deviation.

| Metric | Data Ordering | GPU Non-determinism |
|---|---|---|
| l2 weight difference | $133.2 \pm 9$ | $1.1 \pm 0.07$ |
| l2 output distance | $5.3 \pm 0.03$ | $0.26 \pm 0.02$ |

---

**Algorithm 2** audit

---

INPUT: dataset $D$, epochs $E$, batch size $B$, shared randomness $R$, model $W_\theta$, loss function loss, rounding amount $b_r$, training precision $b_{tr}$, target model precision $b_m$, checkpointing interval $k$, log file $F$ from trainer

OUTPUT: Merkle tree root $M_{root}$

1:  $M_{leaves} \leftarrow$ create empty leaf list
2:  $W_\theta \leftarrow \mathsf{init}(R, b_{tr})$ //`initialize weights`
3:  $T \leftarrow \frac{D*E}{B}$
4:  **for** $t = 1...T$ **do**
5:      $input \leftarrow \mathsf{batch}(R, D, B)$ // `get data batch`
        // `forward pass`
6:      **for** layer $l_\theta \in W_\theta.$layers **do**
7:          $output \leftarrow l_\theta(input)$
8:          **for** $output_i \in output$ **do**
9:              // `Match trainer rounding`
10:             $c \leftarrow \mathsf{read}(output_i, F)$
11:             $output_i \leftarrow \mathsf{rev}(output_i, b_r, c)$
12:         **end for**
13:         $input \leftarrow output$
14:     **end for**
15:     $loss \leftarrow \mathsf{loss}(output)$
16:     // `backward pass`
17:     $grad\_output \leftarrow \nabla_{\mathsf{loss}}$
18:     **for** layer $l_\theta \in W_\theta.$layers **do**
19:         $grad\_input \leftarrow \nabla_{l_\theta}(grad\_output)$
20:         **for** $grad\_input_i \in grad\_input$ **do**
21:             // `Match trainer rounding`
22:             $c \leftarrow \mathsf{read}(grad\_input_i, F)$
23:             $grad\_input_i \leftarrow \mathsf{rev}(grad\_input_i, b_r, c)$
24:         **end for**
25:         $grad\_output \leftarrow grad\_input$
26:     **end for**
27:     $\theta \leftarrow$ update `update weights`
28:     **if** $t \bmod k = 0$ **then**
29:         $M_{leaves}.\mathsf{append}(\mathsf{hash}_{\mathsf{sha256}}(\theta \text{ in precision } b_m))$
30:     **end if**
31: **end for**
32: $M_{root} \leftarrow \mathsf{tree}(M_{leaves})$ // `create Merkle tree`
33: **return** $M_{root}$

---

**Algorithm 3** threshold

INPUT: layer $l$, rounding amount $b_r$, training precision $b_{tr}$
OUTPUT: threshold $\tau$

1: $P \leftarrow$ initialize empty list
2: $N, T \leftarrow$ initialize large # of data points and iterations
3: **for** i=1...N **do**
4:    $GPU1, GPU2 \leftarrow$ select two different GPU architectures
5:    $x \leftarrow$ select random input for layer $l$ in $b_{tr}$ floating-point precision
6:    $y_1 \leftarrow l_{GPU1}(x), y_2 \leftarrow l_{GPU2}(x)$, apply layer $l$ on input $x$ on each GPU
7:    **if** $\mathsf{rnd}_{b_r}(y_1) \neq \mathsf{rnd}_{b_r}(y_2)$ **then**
8:        **if** $y_1 > \mathsf{rnd}_{b_r}(y_1)$ and $y_2 < \mathsf{rnd}_{b_r}(y_2)$ **then**
9:            $P.\mathsf{append}(|y_1 - \mathsf{rnd}_{b_r}(y_1)|)$
10:           $P.\mathsf{append}(|y_2 - \mathsf{rnd}_{b_r}(y_2)|)$
11:       **end if**
12:       **if** $y_1 < \mathsf{rnd}_{b_r}(y_1)$ and $y_2 > \mathsf{rnd}_{b_r}(y_2)$ **then**
13:           $P.\mathsf{append}(|y_1 - \mathsf{rnd}_{b_r}(y_1)|)$
14:           $P.\mathsf{append}(|y_2 - \mathsf{rnd}_{b_r}(y_2)|)$
15:       **end if**
16:   **end if**
17: **end for**
18:  //binary search to select threshold
19: $lower, upper, \tau \leftarrow 0.25 * (2^{-23}), 0.5 * (2^{9-b_r}), 0$
20: **for** t=1...T **do**
21:    $\tau \leftarrow (lower + upper)/2$
22:    $success \leftarrow True$
23:    **for** $p_i \in P$ **do**
24:       exp $\leftarrow$ get exponent of $p_i$
25:       **if** $p_i < \exp * \tau$ **then**
26:           $success \leftarrow False$
27:       **end if**
28:    **end for**
29:    **if** $success$ **then**
30:       $lower \leftarrow \tau$
31:    **else**
32:       $upper \leftarrow \tau$
33:    **end if**
34: **end for**
35: **return** $\tau$

**Algorithm 4** log

INPUT: value $x$, rounding amount $b_r$, threshold $\tau$, file $F$

1: exp $\leftarrow$ get exponent of $x$
2: **if** $|x - \mathsf{rnd}_{b_r}(x)| > \exp * \tau$ **and** $x < \mathsf{rnd}_{b_r}(x)$ **then**
3:    write$(2, F)$ // log rounding up
4: **else if** $|x - \mathsf{rnd}_{b_r}(x)| > \exp * \tau$ **and** $x > \mathsf{rnd}_{b_r}(x)$ **then**
5:    write$(0, F)$ // log rounding down
6: **else**
7:    write$(1, F)$ // log rounding ignore
8: **end if**

