# OpenReview forum: "Optimistic Verifiable Training by Controlling Hardware Nondeterminism"
_NeurIPS.cc/2024/Conference — NeurIPS 2024 poster_

### Official Review · Reviewer_V9Xz · 2024-07-03

**Soundness:** 4
**Presentation:** 4
**Contribution:** 3
**Rating:** 6
**Confidence:** 3

**Summary:**

The authors present a novel approach to achieving identical results in model training across different GPU types and introduce a verifiable training scheme. To achieve this, they:
- proposed a technique for two parties training the same model on different GPU types to achieve identical results by sharing rounding decisions.
- presented a verifiable training scheme which uses a Merkle tree to store model weights for efficient comparison between a trainer and an auditor.
- conducted experiments demonstrating the scalability of the approach to ResNet-50 and GPT-2 across three different NVIDIA GPU architectures (A40, Titan XP, RTX 2080 Ti).
- proposed methods to reduce the storage cost via efficient encoding of rounding logs and an adaptive threshold mechanism to minimize the amount of rounding decisions logged.
- compared their approach with existing methods, including proof-based systems, and the results show that their approach is storage and time-efficient.

**Strengths:**

- The paper introduces a novel method for achieving identical training results across different GPU types by sharing rounding decisions, which could significantly enhance reproducibility in machine learning. The use of a verifiable training scheme based on a well-established verification game adds a layer of trust and transparency to the training process, making it more reliable for sensitive applications. The methods to reduce storage costs via efficient encoding of rounding logs and an adaptive threshold mechanism address practical concerns related to resource usage. This in itself is a huge contribution.
- The experiments using foundation models like ResNet-50 and GPT-2 across multiple GPU architectures showcases the robustness and practicality of the proposed approach. The paper also provides thorough comparisons with existing methods, highlighting the improved storage and time efficiency of the proposed approach, which strengthens the case for its adoption.

**Weaknesses:**

- The authors implemented their verifiable training method entirely on top of the PyTorch framework, using torch version 1.13.1. Given that PyTorch has since released version 2.3.1, there may be compatibility issues or inaccuracies if the method is implemented on the updated version, i.e., using an older version could affect the claim that their method achieves perfect training replication of the two used models, if someone else tries to implement their approach using the updated version, or another framework.
- While the paper includes a comparison with existing methods, the authors assume certain metrics from the baselines due to the unavailability of specific information. This may affect the fairness of the comparisons, especially as the reported improvements might only be valid for the given scenarios (and would be different for other scenarios).

**Questions:**

Please attend to weaknesses above.

**Limitations:**

The authors present the limitation of their work in the limitation section.

---

> ### Author Rebuttal · Authors · 2024-08-07
>
> Thank you for your review! We address concerns regarding pytorch version and fairness comparisons below.
>
> **PyTorch Version**
> We have re-run our experiments with PyTorch 2.3.1, and can confirm our method achieves perfect training replications both within the new version, and between versions. This intuitively makes sense: we do not use any PyTorch methods that would likely change in a way that compromises our method’s effectiveness. Beyond standard methods used for model training, we rely on the following 2 specific PyTorch functions:
>
> 1. torch.nextafter: This returns the next floating point after the given input for a given direction. Since this is a static property of the input’s data type (e.g. 32-bit float), this should not change between versions. Even if it did, this part of our code could be replaced by a hand-written function using knowledge of the IEEE floating point standard.
> 2. module.register_forward_hook : This registers a global forward hook that will be called in every forward pass. This function is widely used in ML interpretability research, as it allows one to examine internal layers of a neural network. We therefore do not expect its core functionality to change in future updates. Even if it did, this part of our code could be replaced by re-writing rounding logic as a model layer.
>
> While we implement our method in PyTorch for convenience, because GPU nondeterminism is fundamentally due to floating point non-associativity, we do not expect updates in PyTorch to compromise our ability to replicate training between GPU devices..
>
>
> **Fairness Comparison**
> Thank you for your feedback, and we agree that some prior works omit information to provide an exact comparison. For this reason, we have been intentionally overly conservative in our comparisons, and believe the reported benefit of our method is a *lower* bound on the actual improvement. Specifically:
>
> 1. Garg. et. al., 2023 do not report the time required for the offline phase of their method. Our method improves on theirs without even including this offline phase. Whether their offline phase takes 0 seconds or more, our method is more time efficient. Furthermore, their work uses techniques specific to logistic regression, so it is not possible to extend their reported time to other training settings.
> 2. Abbaszadeh et. al., 2024 state that they use a simplified version of VGG-11 with fewer parameters. The cost of their method, like ours, would increase with the number of parameters, as that increases the number of computation steps. We report results for the full VGG-11 model, so matching their model would only improve the benefit our method provides.
>
> Unfortunately, there exist very few approaches that scale to large-scale training tasks. However, we strongly believe that the scale of the gains of our method over proof-based alternatives would hold across many settings, as proof-based systems are fundamentally difficult to scale.

---

> > ### Comment · Reviewer_V9Xz · 2024-08-08
> >
> > The authors claimed that they re-run their experiments with PyTorch 2.3.1. I would suggest that they add the results to the paper to further show that their method is framework agnostic. I appreciate that they did further experiments to show this. Kudos!

---

> > > ### Author Response · Authors · 2024-08-12
> > >
> > > Thank you! We will update the paper with our result of achieving training replicability with PyTorch 2.3.1. We would like to note that none of the values in our Figures changed with this update, as our method achieves 0 distance between model weights, and that result stays the same.

---

### Official Review · Reviewer_DqT4 · 2024-07-13

**Soundness:** 3
**Presentation:** 3
**Contribution:** 3
**Rating:** 6
**Confidence:** 3

**Summary:**

Machine learning is increasingly compute power consuming, and as a result, clients may delegate the training to external parties with high computation power. A challenge is how to verify the external parties is training the model as promised.

This work examines a way of auditing by having a trusted third party to verify the outcome. In particular, it investigates the impact of discrepancy in hardware to the third party's auditing outcome.

**Strengths:**

The overall approach of the paper is sound and the assessments are refreshing.

**Weaknesses:**

My main concern of the work is the motivation. In particular, while hardware nondeterminism is a problem, is it really the bottleneck to such auditing? The existence of a trusted third-party is not very obvious to me already: if it exists, we can just delegate the training to this third party; if it doesn't have the compute power to verify all training process of the trainer, then the trainer can still acts suspiciously when not audited in the round.

The setting of the work is hardly convincing to me.

============= After Author Response ==================================

I appreciate the response from the authors. The response cleared some of my doubts over the setting.
Coming from a ML security/privacy background, I still don't buy the existence of a trusted third party. However, I appreciate the contribution of this auditing process as a first step in a non adversarial environment. Therefore, I've raised my score accordingly.

**Questions:**

As mentioned in the weakness box, the setting really confuses me. Please elaborate/justify the scenario in the author response. Thanks!

The technical contribution of mitigating the gap due to hardware difference is interesting. I feel such a technique should have a more impactful and realistic corresponding setting in ML.

**Limitations:**

Limitations are adequately addressed.

---

> ### Author Rebuttal · Authors · 2024-08-07
>
> Thank you for your review! We would like to provide further clarification on the motivation of our work, which arose from real problems faced by industry partners, as the need for robust verification schemes is increasing rapidly (e.g. Together AI, which offers fine-tuning services for open source models, has more than 45,000 registered developers, but no method for these clients to verify that training on their data was performed correctly).
>
>
> Concretely, as described in Section 7, our work makes a 1-out-of-n honesty assumption for verifiable training; i.e., as long as one of n auditors is honest, any attack from a malicious trainer will be detected. This opens two possible settings for our work:
>
> 1. **No one single trusted auditor (de-centralized)**: In this case, a client would outsource model training to many service providers/GPUs, and use them to audit each other. For example, Service A can provide both the model and the corresponding rounding log to a client, perhaps at a premium cost. The client can then approach Service B with this rounding log, and use B to confirm A performed the task correctly, or decide between A and B if the models differ. Different services can now compete on how available these logs are for auditing, and clients can collectively deem services that better enable auditing as more “trustworthy”. This overall mitigates the disadvantage developers who do not have access to vast training resources currently face.
> This problem setting does indeed arise in practice, such as with companies like Gensyn or Ora, which seek to create a large, decentralized compute cluster for AI tasks like model training. In fact, Gensyn currently uses a protocol that’s main limitation is lack of control of hardware non-determinism, which our paper improves [1]. Finally, we note that recent discussions with others in industry (e.g. Atoma Network), have shown interest in applying our method for repeated model inference (e.g. text generation), where GPU nondeterminism leads to divergent outputs. Therefore, there does exist strong, real-world motivation for controlling hardware non-determinism between two parties for machine learning tasks.
>
> 2. **Trusted 3rd party auditor**: This setting motivates the existing academic literature our work is based on. For example, the proof-of-learning protocol in [2] describes a setting where a legal entity serves as a verifier, and re-performs gradient computations at specific steps to verify training. Legal entities may not have the availability to provide model training as a large-scale service, but could provide auditing services at a cost to the client. In these cases, the client may choose to pursue auditing only in serious cases where they are highly suspicious of the model they received, and wish to pursue dispute resolution. Even if auditing is not performed for all training tasks, our method would allow model training service providers to build a reputation for passing/failing trusted 3rd party audits, increasing overall trust in the rapidly growing model-training-as-a-service space.
>
> While we chose the framing of an “auditor” to stay close to related academic works we build on, we agree with the reviewer that the broader motivation and existence of the de-centralized setting could be made clearer. We will therefore make the following additions to the paper:
>
> 1. Line 34 in Introduction: An alternative ``optimistic'' approach is to consider a 3rd party verifier. This could be a trusted 3rd party,  such as a non-profit organization that may not have sufficient computing resources to provide model training as a service beyond auditing, or a different service provider that the client approaches and wishes to compare with the original trainer.
> 2. Line 44 in Introduction: Applying verifiable computation techniques to the setting of model training is particularly important given the increase in decentralized machine learning services like Gensyn, which seek to make ML compute more accessible by creating a network of many untrusted GPUs.
> 3. Replace mentions of “trusted 3rd party auditor” to “3rd party (e.g. trusted auditor or other service provider)”
>
>
> [1] https://docs.gensyn.ai/litepaper
> [2] https://arxiv.org/pdf/2103.05633

---

> > ### Comment · Reviewer_DqT4 · 2024-08-12
> > **Thanks for the response**
> >
> > I've updated my score accordingly.

---

### Official Review · Reviewer_RB9A · 2024-07-15

**Soundness:** 3
**Presentation:** 3
**Contribution:** 3
**Rating:** 7
**Confidence:** 1

**Summary:**

This paper studies the problem of verifying the correctness of the training process of model. In particular, the user who lacks sufficient resources pays a service provider with sufficient resources to train models. Then a trusted third-party auditor will check whether the training process is legit. The proposed approach rounds after intermediate computation steps, and stores rounding decisions based on an adaptive thresholding procedure, to successfully control for nondeterminism among different GPUs and also saves the storage space. Empirical results show that the proposed approach can run more efficiently than the previous baselines while also using smaller storage space, which makes it valuable for verifying larger models.

**Strengths:**

1. The problem of verifying the correctness of the model training process is an important problem and leveraging a third-party auditor can be one possible way to scale these techniques to larger models.
2. The proposed approach outperforms the existing baselines both in terms of verification efficiency and usage of storage space.

**Weaknesses:**

There is a lack of description on the relation and difference of this work and Teutsch & Reitwießner (2019). I think the authors will benefit by clarifying the difference and as well as emphasizing their novelty.

**Questions:**

There is no direct outstanding questions, but I would encourage the authors to focus on the writing to highlight the contribution more explicitly.

**Limitations:**

Limitations are well addressed in this work.

---

> ### Author Rebuttal · Authors · 2024-08-07
>
> Thank you for your review! Our work is the first to show how to apply Teutsch & Reitwießner (2019)’s method, which was developed specifically for blockchain verification, to a machine learning setting. As such, our method needs to address GPU nondeterminism challenges that Teutsch & Reitwießner (2019) do not need to. Concrete differences includes:
>
> 1. **Novel Setting**: The motivation of Teutsch & Reitwießner (2019) is to address the “verifier’s dilemma” that occurs in blockchain protocols where verifying a transaction (which ensures the security of the blockchain) is too computationally intensive for a user, resulting in less financial incentive for blockchain “miners” to perform verification. They do not explore machine learning applications, including our goal of verifiable training, where the outsourced computation needs to take place on specialized hardware like GPUs.
> 2. **Rounding to Eliminate Nondeterminism**:  Because the protocol in Teutsch & Reitwießner (2019) does not need to address nondeterminism issues, their approach of both parties independently storing intermediate computation outputs in a Merkle tree suffices.  In our work, both parties need to round intermediate computations and share rounding directions, as described in Section 5.3. This modifies the computation task and is a more interactive procedure.
> 3. **Rounding Log Storage**:   Unlike Teutsch & Reitwießner (2019), our method requires parties to store rounding log information to address nondeterminism. Our paper therefore proposes a novel method for reducing the storage cost, as described in Section 5.4, that describes an adaptive threshold selection procedure unique to our setting.
>
> To address the reviewer’s concerns, we will add the following sentences in line 66 in the Introduction:
>
> *"We show how to adapt the verification game from Teutsch & Reitwießner (2019), which that was proposed for verifying transactions in blockchain protocols, where an efficient Merkle tree data structure stores intermediate computation outputs for computationally intensive tasks. Unfortunately, naively adapting this strategy for verifiable training, where intermediate computation outputs are model checkpoints, fails due to hardware nondeterminism, as the root hashes of the Merkle trees will not match even when all parties are honest. We therefore demonstrate a rounding-based procedure (Section 5.3), as well as a method for reducing its storage cost (Section 5.4), to successfully eliminate nondeterminism challenges. We additionally show our method outperforms baselines for verifying model training."*

---

> > ### Comment · Reviewer_RB9A · 2024-08-12
> >
> > Thanks for the response. Please incorporate the promised changes in the final version.

---

### Author Rebuttal · Authors · 2024-08-07

We would like to thank all reviewers for their valuable feedback to improve our work. Reviewers found our work addresses an important problem (RB9A), our technical contribution interesting (DQT4), and that our proposed method is both robust and practical in the context of large-scale foundation models (V9XZ).

We address each reviewer’s specific concerns in the individual responses.

---

### Decision · Program_Chairs · 2024-09-25

**Decision:**

Accept (poster)

**Comment:**

This work proposes an algorithm for verifying the correct training of a machine learning model in a setting of this training be delegated to an external service provider. This work uses several techniques to reduce computation, control for hardware nondeterminism, and save storage cost.

Overall, there is consensus that this work provides a novel contribution that is sound and improves over a well chosen set of benchmarks. There is some limitation in the practicality of this approach in that this work makes the strong assumption of a trusted third party. However, given that this is an assumption that has been used several times before, it is not unfounded.